# Optimization of Encapsulation by Ionic Gelation Technique of Cryoconcentrated Solution: A Response Surface Methodology and Evaluation of Physicochemical Characteristics Study

**DOI:** 10.3390/polym14051031

**Published:** 2022-03-04

**Authors:** María Guerra-Valle, Guillermo Petzold, Patricio Orellana-Palma

**Affiliations:** 1Departamento de Nutrición y Dietética, Facultad de Ciencias para el Cuidado de la Salud, Campus Concepción, Universidad San Sebastián, Lientur 1457, Concepción 4080871, Chile; maria.guerra@uss.cl; 2Laboratorio de Crioconcentración, Departamento de Ingeniería en Alimentos, Facultad de Ciencias de la Salud y de los Alimentos, Campus Fernando May, Universidad del Bío-Bío, Av. Andrés Bello 720, Chillán 3780000, Chile; 3Departamento de Ingeniería en Alimentos, Facultad de Ingeniería, Campus Andrés Bello, Universidad de La Serena, Av. Raúl Bitrán 1305, La Serena 1720010, Chile

**Keywords:** encapsulation, hydrogel, beads, cryoconcentration, model solution

## Abstract

The objective of this study was to evaluate the optimal conditions to encapsulate cryoconcentrate solutions via ionic gelation technique. Hydrogel beads were prepared using alginate (1%, 2% and 3% (*w*/*w*)) and cornstarch (0.5%, 1% and 2% (*w*/*w*)). Later, a sucrose/acid gallic solution was concentrated through block freeze concentration (BFC) at three cycles. Thus, each solution was a mixture with the respective combination of alginate/cornstarch. The final solution was added drop-wise on a CaCl_2_ solution, allowing the formation of calcium alginate-cornstarch hydrogel beads filled with sucrose/acid gallic solution or cryoconcentrated solution. The results showed that alginate at 2% (*w*/*w*) and cornstarch at 2% (*w*/*w*) had the best efficiency to encapsulate any solution, with values close to 63.3%, 90.2%, 97.7%, and 75.1%, and particle sizes of approximately 3.09, 2.82, 2.73, and 2.64 mm, for initial solution, cycle 1, cycle 2, and cycle 3, respectively. Moreover, all the samples presented spherical shape. Therefore, the appropriate content of alginate and cornstarch allows for increasing the amount of model cryoconcentrated solution inside of the hydrogel beads. Furthermore, the physicochemical and morphological characteristics of hydrogel beads can be focused for future food and/or pharmaceutical applications, utilizing juice or extract concentrated by BFC as the solution encapsulated.

## 1. Introduction

Recently, many studies have shown the potential of block freeze concentration (BFC) to improve different physicochemical properties and bioactive compounds from fruit juices, such as total soluble solid content [1], color [2], polyphenols [3], anthocyanins [4], and flavonols [5], and antioxidant activity [6]. Precisely, in the BFC process, the liquid solution is completely exposed to freezing temperatures, and as the temperature decreases below the freezing point, the water turns into ice crystals (ice fraction), and in turn, the unfrozen fraction (cryoconcentrated solution) remains between the ice crystals by a counter-diffusion phenomenon, and thus, once the freezing stage has finished, the cryoconcentrated solution can be extracted from the ice fraction by gravitational methods (natural thawing) [7] or by external forces such as centrifugation [8], vacuum [9], or centrifugation-filter [10]. Thereby, the cryoconcentrated solution has presented endless improvements in terms of quality properties in comparison to the original fresh juice [11].

Despite all the benefits and interesting properties in the cryoconcentrated juices, there are limitations on their properties in terms of stability, bioactivity, and bioavailability, due to factors such as processing (cooking, germination, extrusion, and/or fermentation, among others), environmental features (temperature, pH, light, and oxygen levels), storage time [3], and even the effects under gastrointestinal conditions, i.e., the fresh and/or cryoconcentrated juices, can be affected by internal or external conditions [12]. Therefore, it is important to protect the cryoconcentrated solutions against different factors through a simple, versatile, practical, and economical technology, avoiding the color degradation and the progressive loss of nutritional properties.

Hence, in the last few decades, the encapsulation by ionic gelation has emerged as an effective technology to contain liquid samples in biopolymer matrices due to the low temperatures involved in the encapsulation process, and thus, this method allows minimal modification of the quality properties in the encapsulated solution [13]. Specifically, the ionic gelation method corresponds to the interaction between polymers with oppositely charged polymers or a polymer with a polycation or polyanion [14]. Commonly, alginate solution (anionic polymer), a natural polysaccharide, has been used as an encapsulating material due to its low cost, non-toxic characteristics, and high biocompatibility with other solutions. Thus, alginate solution is extruded drop by drop with a needle into a calcium chloride solution, conforming a crosslinked structure due to the formation of ionic bonds for the interaction with the polyvalent cations of the gelation medium (Ca^2+^), allowing the construction of a polymeric microparticle, also called hydrogel beads [15]. Whereby, the hydrogel beads obtained by ionic gelation have been widely used for the immobilization of different food solution, protecting important bioactive compounds, such as betacyanins [16], betaxanthin [17], and ellagic acid, quercetin, kaempferol, and caffeic acid [18], anthocyanins [19,20], and polyphenols [21]. However, to the best of our knowledge, the encapsulation of any cryoconcentrated solution by ionic gelation technique has not been reported.

Additionally, the use of an experimental design (DOE) allows one or more variables (factors) to be deliberately manipulated to determine the effects on a variable of interest (experimental response) [22]. Specifically, the response surface methodology (RSM) allows for the obtainment of optimal conditions in many areas of knowledge. In this context, the literature exhibits extensive applications of optimization processes, for example, in the search of the best conditions in the non-isothermal crystallization of isotactic Polypropylene (iPP) [23], in the optimization of annealed HSLA Steel [24], in the search of eco-friendly flame retardant finish for cotton fabric [25], and in the optimization of organophosphorus fire-resistant finish for cotton fabric [26].

Thus, the objective of this work was to investigate the effects of encapsulation of cryoconcentrated solution (sucrose/acid gallic mixture) obtained by BFC through the ionic gelation method to form stable calcium alginate hydrogel beads filled with cryoconcentrated solution. Additionally, total soluble solid content, gallic acid content, particle size, shape, efficiency of encapsulation, loading capacity, water activity, moisture content, and bulk density were determined. Hence, the innovation of this study is the first step on encapsulation of cryoconcentrated model solution for future formation of calcium alginate hydrogel beads filled with cryoconcentrated samples obtained from different liquid foods, such as fruit juices or plant extracts. The results provided information for the best alternative in terms of alginate and cornstarch concentration to encapsulate cryoconcentrate solutions.

## 2. Materials and Methods

### 2.1. Chemicals and Reagents

Sodium alginate, sucrose, ethanol, calcium chloride (CaCl_2_), sodium carbonate (Na_2_CO_3_), Folin–Ciocalteu reagent, and gallic acid (GA) were supplied from Merck (Hohenburn, Germany). Distilled water was used throughout. All of the chemical products have high purity grade (99.9%) which were used in this study. Commercial corn starch was purchased from a local market (Maizena^®^, Unilever, Santiago, Chile).

### 2.2. Preparation of Cryoconcentrated Solutions

Firstly, the initial model solution was prepared from the results obtained by Guerra-Valle et al. [27], where the total soluble solid content (TSSC) content was close to 15 °Brix and gallic acid (GA) content was approximately 1000 mg/L. Later, the model solution was subjected to the BFC procedure indicated by Orellana-Palma et al. [5]. Specifically, 45 mL of sucrose/GA mixture was deposited in centrifuge tubes, and then, the tubes were covered with thermal-insulating foam, where only the upper part of the tubes was exposed to the freezing front, and thus, the freezing step (overnight at −20 °C) was produced by axial direction in a vertical static freezer (280, M and S Consul, Sao Paulo, Brazil). Once the freezing step ended, the cryoconcentrated solution was extracted from the frozen fraction using a centrifuge (Eppendorf 5430R, Hamburg, Germany) at 15 °C for 20 min at 1600 RCF (4000 rpm). The BFC procedure was applied at three cycles, i.e., the cryoconcentrated fraction from the first extraction (cycle 1) was used as a feed solution for the subsequent cycles using the same procedure described above (axial freezing at −20 °C and centrifugation at 15 °C for 20 min at 1600 RCF (4000 rpm)). The BFC procedure at three cycles is shown in Figure 1. In each cycle, TSS content and GA content were determined. Thereby, the initial sample, cycle 1, cycle 2, and cycle 3 presented values close to 15.0 °Brix and 1000 mg GAE/L, 27.7 °Brix and 2117 mg GAE/L, 40.7 °Brix and 8309 mg GAE/L, and 49.5 °Brix and 11,100 mg GAE/L, respectively.

The reason for an axial freezing is as follows. The FC is based on the movement of solutes during the formation of ice crystals during the freezing process and due to this process, the cryoconcentrated solution is expelled and accumulated in the liquid fraction (between ice crystals). Therefore, it is very important to control the freezing rate, since, when the freezing rate was higher than 8 μm/s, the freezing occurred too rapidly to expect a considerable separation of the concentrated solution from the ice crystals, i.e., the ice occluded solutes during the freezing process [28]. Thereby, from previous studies, the axial freezing presented a lower freezing rate than radial freezing, since radial treatments have freezing front propagation in all radial directions, while axial treatments have freezing front propagation in only a one-way direction axis. Hence, the axial freezing process is slower, and it allows for moderate crystal growth with better separation of cryoconcentrate solution from the water (ice crystals), i.e., the movement by counter-diffusion of solutes move away from the growing crystal surface and accumulate at the solid-liquid interface and was better in axial freezing than radial freezing [29,30].

### 2.3. Encapsulation of Cryoconcentrated Solutions

The encapsulation of initial sucrose/GA solutions and cryoconcentrated samples was prepared by the ionic gelation method as described by Fathordoobady et al. [31], with modifications. Firstly, sodium alginate/cornstarch powder was dissolved into distilled water with constant agitation (overnight) using a magnetic stirrer (RSM-14 HP, ProfiLab24 GmbH, Berlin, Germany) at 40 °C. Moreover, CaCl_2_ powder was dissolved into distilled water with constant agitation using a magnetic stirrer, preparing a concentration of 0.5 M. Thus, the sodium alginate/cornstarch solution was mixed with a sucrose/GA solution (initial or cryoconcentrated) under mild agitation (250 rpm) for 30 min. Once homogenized, each solution was added drop-wise on the CaCl_2_ solution (in constant mild agitation with a magnetic stirrer) through a fine stainless-steel needle (internal diameter tip: 0.8 mm), with a height between the tip and the surface of the CaCl_2_ solution close to 15 cm. Then, the beads were maintained in the gelling bath (CaCl_2_) for 30 min, and later, the beads were filtered and subsequently washed with distilled water -.

### 2.4. TSSC

The TSSC in the initial sucrose/GA solution and cryoconcentrated samples was determined using a handheld refractometer (Pocket Pal-1, Atago Co., Ltd., Tokyo, Japan) and the results were expressed as °Brix. Distilled water was used as the calibration substance.

### 2.5. GA Content

The GA content in the initial solution, cryoconcentration samples, and beads was quantified by the Folin–Ciocalteu assay [32], with slight modifications. Specifically, 100 μL of solution was mixed with 7900 μL of deionized water, and then, 500 μL of Folin–Ciocalteu reagent was added. After 5 min, 1500 μL of 7.5% Na_2_CO_3_ solution was added to the mixture. Thus, the mixture was allowed to stand in the dark at room temperature for 90 min (incubation), and later, the absorbance at 760 nm was recorded, using gallic acid as the standard. The samples were measured using a UV–vis spectrophotometer (T70, Oasis Scientific Inc., Greenville, SC, USA). The results were expressed as mg GA equivalents (GAE) per liter of sample (mg GAE/L).

### 2.6. Parameters of Encapsulation

#### 2.6.1. Efficiency of Encapsulation (EE%)

The beads were deposited in a mortar and crushed with a pestle [14]. Thereby, the EE% is the GA concentration encapsulated into the beads with respect to the initial GA concentration. Thus, the EE% was calculated according to Equation (1).
(1)$$\mathrm{EE}\left(\%\right)=\;\left(\frac{TBC_{c}}{TBC_{0}}\right)\times 100\%$$
where *TBC_c_* is the GA concentration released from the beads, and *TBC*_0_ is the respective initial GA concentration added during the encapsulation process (initial or cryoconcentrated solutions).

#### 2.6.2. Loading Capacity (LC%)

The beads were dried in an oven at 120 °C for 2 h [33]. Thereby, the LC% was calculated according to Equation (2), where the LC% is the relation between the weight of the beads after being dried and the weight of the beads prior to being dried.
(2)$$\mathrm{LC}\left(\%\right)=\;\left(\frac{W_{0}-W_{f}}{W_{0}}\right)\times 100\%$$
where *W*_0_ is the weight of the beads before the drying process, and *W_f_* is the weight of the beads after the drying process.

### 2.7. Characterization of Calcium Alginate Hydrogel Beads Filled with Cryoconcentrated Solution

#### 2.7.1. Particle Size

The particle size of the beads was measured by digital image analysis (1600 × 1200 pixels) using ImageJ^®^ software [34], where fifty beads were selected per microphotograph. The microphotographs were obtained in an Olympus trinocular microscope (BX51, Olympus Co., Tokyo, Japan) equipped with an Olympus digital camera (LC20, Olympus Co., Tokyo, Japan).

#### 2.7.2. Shape

The shape was quantified to identify the roundness of the beads using the sphericity factor (SF) [35] through microphotographs in an Olympus trinocular microscope (BX51, Olympus Co., Tokyo, Japan) equipped with an Olympus digital camera (LC20, Olympus Co., Tokyo, Japan), where SF equal to zero specifies a symmetrical sphere, while higher values indicate a greater amount of distorted bead; SF was calculated according to Equation (3).
(3)$$\mathrm{SF}=\frac{d_{max}-d_{min}}{d_{max}+d_{min}}\;$$
where *d_max_* is the largest diameter passing through the bead centroid and *d_min_* is the smallest diameter perpendicular to *d_max_*.

#### 2.7.3. Water Activity (a_w_)

The a_w_ of the samples was determined using a dew-point hygrometer (Aqua Lab Model 4TE, Pullman, WA, USA) at ambient temperature.

#### 2.7.4. Moisture Content

The moisture content was carried out according to the method defined by Kaderides and Goula [36]. The samples were dried in a vacuum oven (3618-1CE, Lab. Line Instruments Inc., Melrose Park, IL, USA) at 70 °C, with the measurement of the weight taken every 2 h. Thus, the process was completed until a constant weight was reached (variation between weight measurements less than 0.3%).

#### 2.7.5. Bulk Density (*ρ*_B_)

The *ρ*_B_ was obtained according to the method described by Abdullah and Geldart [37]. Specifically, a known mass (g) of beads was deposited into a volumetric probe, and later, the volumetric probe was deposited on an analytical balance (Ohaus PX124 AM, TEquipment Inc., Long Branch, NJ, USA). The *ρ*_B_ was calculated by dividing the mass of the beads by the volume occupied in the probe [9].

### 2.8. Experimental Design

The experiments were conducted according to the experimental design given in Table 1. A randomized factorial design 3^2^, considering 2 factors at 3 levels (for each factor) was used to study the optimum combination of alginate and cornstarch solutions in the encapsulation of cryoconcentrated solution through the ionic gelation technique. The levels of independent variables used for the experiment were 1%, 2%, and 3% (*w*/*w*) in alginate solution and 0.5%, 1%, and 2% (*w*/*w*) in cornstarch solution. A total of nine experiments were performed separately for getting the experimental response for the dependent variables, such as particle size, shape, EE%, a_w_, moisture content, and bulk density.

### 2.9. Statistical Analysis

All the experiments were conducted in triplicate, and the results were reported as mean ± standard deviation. Results were subject to analysis of variance (ANOVA) and least significant difference test (LSD) using Statgraphics Centurion XVI Software (Statpoint Technologies Inc., Warrenton, VG, USA) to obtain the statistical analysis of the data of the experimental design (*p* ≤ 0.05) and graphical analysis of the data (response surface methodology (RSM) and contour plots).

## 3. Results and Discussion

### 3.1. Efficiency of Encapsulation (EE%)

Table 2 shows the behavior of the mixture in terms of EE% for the initial solution (Figure 2a), cycle 1 (Figure 2b), cycle 2 (Figure 2c), and cycle 3 (Figure 2d).

Firstly, for all the cases, the minimum concentration of alginate (1% *w*/*w*) and cornstarch (0.5% *w*/*w*) presented the lowest EE% values, with 12.5%, 37.9%, 59.4%, and 19.3%, for initial solution, cycle 1, cycle 2, and cycle 3, respectively, but an increase in the concentration of cornstarch (with alginate (1% *w*/*w*)) allowed a significate increase in the EE%, since the values were close to 21.3% and 25.8% for initial solution, 60.8% and 64.4% for cycle 1, 64.6% and 69.0% for cycle 2, and 22.1% and 32.2% for cycle 3, with cornstarch at 1% and 2% (*w*/*w*), respectively. Thus, alginate at 1% (*w*/*w*) with any concentration of cornstarch and encapsulated solution presented lower EE% values than the other alginate/cornstarch combinations. Thereby, a gradual increase in the concentrations of alginate and cornstarch indicated a significant increase in EE%, since alginate at 2% and 3% (*w*/*w*) presented EE% values ranged from 47.7% to 63.3%, and 39.0% to 57.2%, for initial solution, 75.9% to 90.2%, and 52.1% to 70.2%, for cycle 1, 82.9% to 97.7%, and 64.9% to 83.5%, for cycle 2, and 50.7% to 75.1%, and 40.0% to 53.6%, for cycle 3, from 0.5% to 2.0% (*w*/*w*) of cornstarch, respectively. Hence, the encapsulation process (with any concentration of alginate/cornstarch) with cryoconcentrated solution from cycle 2 presented the maximum EE% values in comparison to the initial solution, cycle 1, and cycle 3.

Additionally, for a complete visualization, Figure 2 shows the RSM surface in terms of EE% for the initial solution (Figure 2a), cycle 1 (Figure 2b), cycle 2 (Figure 2c), and cycle 3 (Figure 2d).

Then, from the EE% values, the best combination to encapsulate any solution (from initial solution to cycle 3) in comparison to the other combinations was alginate at 2% (*w*/*w*) and cornstarch at 2% (*w*/*w*), since the EE% values were close to 63.3% for initial solution, 90.2% for cycle 1, 97.7% for cycle 2, and 75.1% for cycle 3, respectively. Moreover, alginate at 3% (*w*/*w*) with any concentration of cornstarch and solution encapsulated had EE% values lower than those of alginate at 2% (*w*/*w*) with the respective concentration of cornstarch and solution encapsulated.

Therefore, by considering all the results from this experimental design, the alginate at 2% (*w*/*w*) with cornstarch at 2% (*w*/*w*) was chosen as the optimal combination to encapsulate the initial solution and each cycle of the BFC process, since these values present the maximum EE% values.

The high EE% values for the selected combinations can be explicated due to the fact that the concentration of alginate at 2% (*w*/*w*) allows for better interlacing between the molecules than other concentrations, forming thicker connections, and thus, this phenomenon strengthens the hydrogel bead matrix, and in turn, reduces the size of the empty spaces (pores), decreasing the output of liquid encapsulated to the outside [38]. Furthermore, the addition of cornstarch allows for filling the pores, increasing the EE% values [39]. Moreover, a concentration of alginate lower than 2% (*w*/*w*) produces hydrogel beads with weak connections, making it more prone to liquid leakage during the encapsulation process, and also, a concentration of cornstarch lower than 2% (*w*/*w*) is not enough to fill the pores generated by the cross-linking of the molecules that form the microsphere. In the same way, cryoconcentrate from the third cycle, alginate higher than 2% (*w*/*w*), and cornstarch higher than 2% (*w*/*w*) affects the properties of the mixture, since the mixture provokes a solution with high viscosity, causing difficult manipulation (extrusion) of the material encapsulated [40]. Besides, the cryoconcentrated solution from cycle 3 presented lower EE% values than those obtained using cycle 2 with any alginate/cornstarch mixture, and this effect can be related to the high viscosity of the cryoconcentrated samples, since studies on the cryoconcentration of liquid samples indicate that the increase in cycles causes a significant loss of water, and in turn, this consistent increase in solutes generates an increase in viscosity properties [5,27], and it provokes a high difficulty in the formation of calcium alginate-cornstarch hydrogel beads with a spherical shape, leading to the formation of capsules with “tails” and irregular shapes [33] (Appendix A).

The EE% values (alginate at 2% (*w*/*w*) and cornstarch at 2% (*w*/*w*)) were higher than in the encapsulation through calcium-alginate (3% (*w*/*w*)) capsules containing cornstarch at 2% (*w*/*w*) of yerba mate (*Ilex paraguariensis*) extract [39], where the EE% was close to 65%. Thus, the EE% was increased with the addition of the “filler”, since it prevents the exit of liquid sample to the external environment, and in addition, depending on the sample (model solution, juice, or extract, among others), the first stage to obtain a good encapsulation is to identify the correct concentration of alginate solution and cornstarch. Furthermore, our high values (in the cycles) can be related to the encapsulated sample, since as the cycles progressed, the viscosity of the sucrose/GA solution increases significantly due to the increase in solids [41], and thus, the cryoconcentrated solution could participate as a “glue”, and it can be easily adhered to inaccessible places (nanopores) for cornstarch. Whereby, for a correct encapsulation (reflected in the efficiency), it is important to clarify the properties of the solutions (alginate solution, cornstarch solution, and encapsulated material), collected distance and the drop size [42]. Thus, a correct concentration of sodium alginate aqueous solutions, divalent cations solution, and pH medium allow for the obtainment of solid and defined calcium alginate hydrogel beads [40], and in turn, the cornstarch covers the pores of the beads due to its respective amorphous, granular (range in size from 1 to 100 μm), and crystalline regions [43].

### 3.2. Loading Capacity (LC%)

Figure 3 shows the LC% for the initial solution, cycle 1, cycle 2, and cycle 3, with the combination of alginate at 2% (*w*/*w*) and cornstarch at 2% (*w*/*w*).

Specifically, statistically significant differences were detected when compared to the hydrogel beads samples (initial solution and cycles). Thereby, from initial solution to cycle 3, the LC% values had descending behavior, with 86.0%, 81.1%, 74.5%, and 68.3% for initial solution, cycle 1, cycle 2, and cycle 3, respectively. This performance displays a marked inverse relationship in relation to the efficiency (until cycle 2), since LC% have a very close connection to the water-soluble characteristic of the solution used (alginate/cornstarch with sucrose/GA solution from initial solution, cycle 1, cycle 2, or cycle 3). Therefore, the sucrose/GA solution encapsulated affects the water-soluble characteristic due to the progressive increase in the viscosity [30], helping in the first step (from initial solution to cycle 2), but later, this important characteristic counteracted the interaction between samples to form the final matrix.

The LC% results were lower than in previous studies in our laboratory, since Petzold et al. [33] achieved LC% values close to 96.6%, where the study was focused in the encapsulation of liquid smoke through calcium alginate hydrogel beads via the ionic gelation technique. Similarly, Deladino et al. [44] obtained values higher than 85% in calcium alginate hydrogels beads (with the same encapsulation process) loaded with yerba mate extract. Hence, as above mentioned, as the cycle progressed, the viscosity of the encapsulated sample increased progressively due to the high TSSC concentration, and thus, these factors affect the solubility of the combination alginate/cornstarch and solution encapsulated prior to the encapsulation process, reflecting notably in cycle 3 (the highest TSSC and viscosity values), since it presents the lowest LC% value.

### 3.3. Characterization of Calcium Alginate Hydrogel Beads

Table 3 shows the characterization of calcium alginate hydrogel beads for the initial solution, cycle 1, cycle 2, and cycle 3, with alginate at 2% (*w*/*w*) and cornstarch at 2% (*w*/*w*) in terms of particle size, SF, a_w_, moisture content, and *ρ*_B_.

Firstly, the size of the hydrogel beads decreased significantly from 3.09 (initial solution) mm to 2.64 mm (cycle 3). This progressive decrease can be related to the interaction of the solution with the alginate/cornstarch mixture, since the initial solution, despite its possible high solubility with the alginate/cornstarch mixture (reinforced by the LC% value), does not have a substantial influence in the molecular bond interactions to close interstitial empty spaces of the gel matrix [45]. Thereby, the final structure (hydrogel beads with initial solution) has a higher distance between pores, generating a larger microscopic structure than hydrogel beads with any cryoconcentrated solution [46]. Specifically, the particle size values (in all cases) present similar values than other studies by ionic gelation technique. Deladino et al. [44] showed an average diameter of around 2.0–2.5 mm for calcium alginate-starch hydrogel beads, with alginate/yerba mate extract at 3% (*w*/*w*) and cornstarch at 2% (*w*/*w*), and thus, the alginate/cornstarch/solution mixture plays a primary role in the size of the pores, and it can be extrapolated on the particle size of the hydrogel beads.

For SF, there are no significant differences between samples, since the SF values were close to 0.02, indicating that the hydrogel beads filled with any solution can be considered spherical, since Chan et al. [47] suggested that hydrogel beads with SF values less than 0.05 have a high degree of sphericity, and thus, a SF value of 0 is a perfect sphere, while values from 0.05 to 1 may lead to an oval or elongated shape [48]. Thus, our results are visually consistent, since, as observed in Figure 4, all the samples with alginate at 2% (*w*/*w*) and cornstarch at 2% (*w*/*w*) presented a spherical shape. A similar behavior was reported by Rohman et al. [49], who encapsulated *Rhodopseudomonas palustris* KTSSR54 using alginate/cornstarch beads through the ionic gelation technique, since the authors indicated that a cornstarch concentration between 3% (*w*/*v*) to 4% (*w*/*v*), with alginate at 2% (*w*/*v*), allows for good sphericity (SF < 0.05) in the alginate beads.

For a_w_ and moisture content, there are significant differences between the samples, with a continuous decrease from the initial solution until cycle 3, with values from 0.98 to 0.92, and from 86.1% to 73.6%, respectively. This progressive decrease is related to the loss of water by syneresis due to the addition of the cryoconcentrated solution since a solution with a high TSSC produces a significant diffusion of water molecules in the hydrogel beads’ formation [39]. Hence, the values obtained are comparable to the data reported by Zazzali et al. [50], since the study described as variables, such as pH, extrusion tip size, and washing/storage protocol, change the structural properties of hydrogel beads, causing a 10% decrease in the a_w_ and moisture content of the samples.

For *ρ*_B_, there is a significant increase in the values as the cycles increased, from 1.04 (initial solution) to 1.16 g/mL (cycle 3). This behavior can be related to the addition of solution encapsulated in the alginate/cornstarch mixture, since the cryoconcentrate from cycle 3 has a greater mass than the previous cycles (and initial solution) due to the high TSSC value (initial solution = 15.0 °Brix, cycle 1 = 27.7 °Brix, cycle 2 = 40.7 °Brix, and cycle 3 = 49.5 °Brix). Thus, a solution treated by BFC has a considerable increase in its density as the cycles increase, affecting the density of the alginate/cornstarch/solution mixture [51]. In the same way, Chan et al. [52] showed a similar phenomenon with respect to the *ρ*_B_, since the authors used different concentrations of cornstarch (0–600 g/L) to reduce the porosity of alginate hydrogel beads, and in turn, the results showed that the *ρ*_B_ of the capsules increased progressively as the cornstarch concentration increased.

## 4. Conclusions

The optimization process by RSM indicated that alginate at 2% (*w*/*w*) and cornstarch at 2% (*w*/*w*) allowed for better efficiency of encapsulation of sucrose/acid gallic concentrated by block freeze concentration assisted by centrifugation than other alginate/cornstarch mixtures, where cornstarch filled the empty space (pores) of the alginate beads. This mixture produced a more suitable matrix (calcium alginate-cornstarch hydrogel beads), and in addition, it increased the amount of model cryoconcentrated solution inside of the hydrogel beads, and in turn, it permitted final beads with spherical shape. Therefore, the results obtained in this study could be used as a first step to develop good sustained-release devices and the possible uses on gastrointestinal conditions or storage situations. Moreover, the hydrogel beads will be filled with juice or extract concentrated by BFC for future food, engineering, and/or pharmaceutical applications.

## Figures and Tables

**Figure 1 polymers-14-01031-f001:**
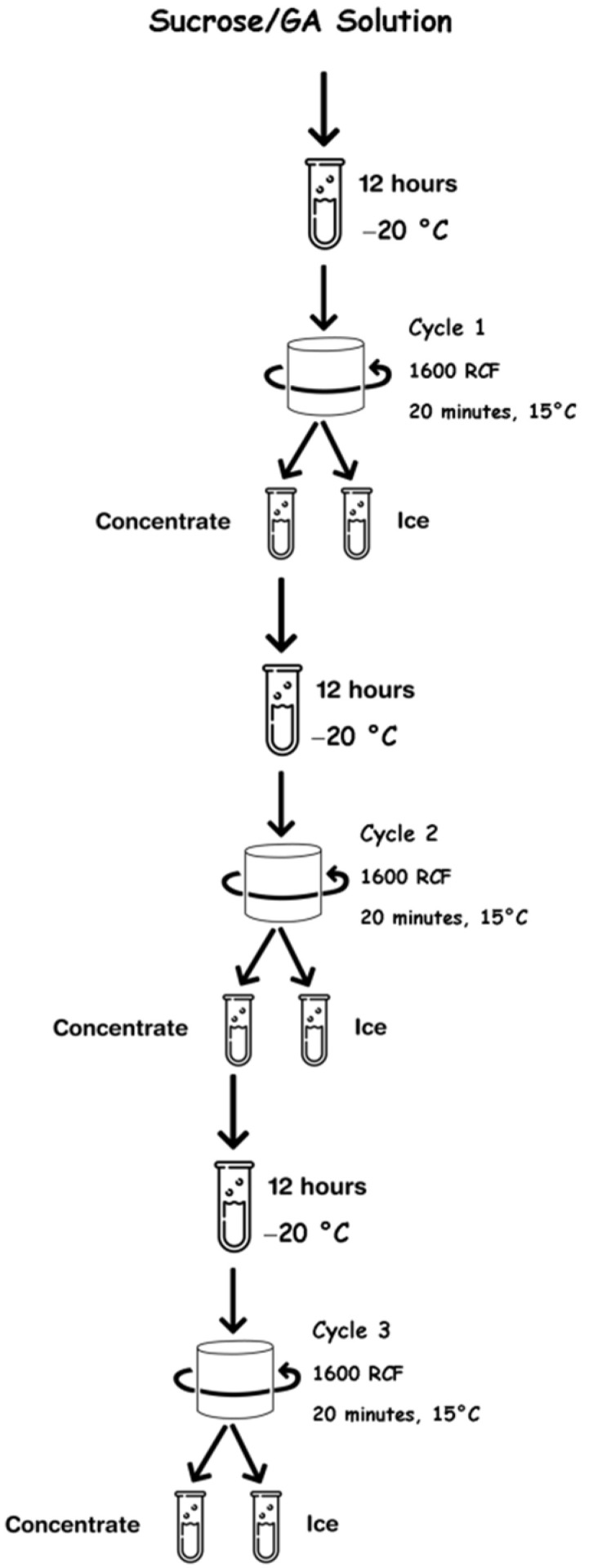
Scheme of BFC process.

**Figure 2 polymers-14-01031-f002:**
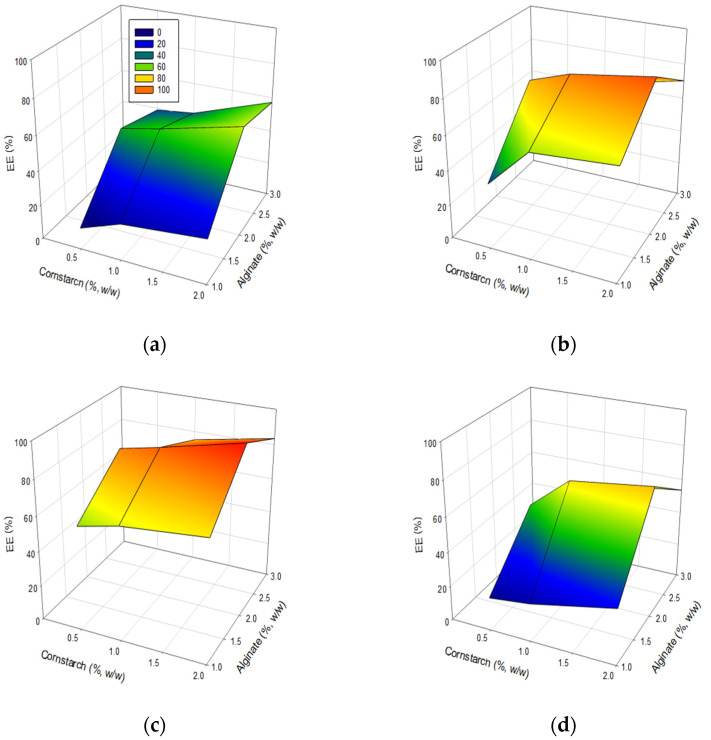
Efficiency of encapsulation: (**a**) Initial solution; (**b**) Cycle 1; (**c**) Cycle 2; (**d**) Cycle 3.

**Figure 3 polymers-14-01031-f003:**
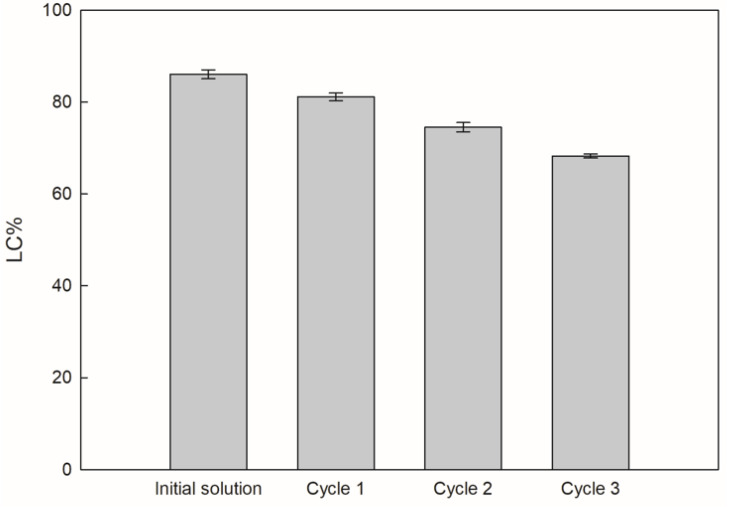
Loading capacity (LC%) in alginate at 2% (*w*/*w*) and cornstarch at 2% (*w*/*w*).

**Figure 4 polymers-14-01031-f004:**
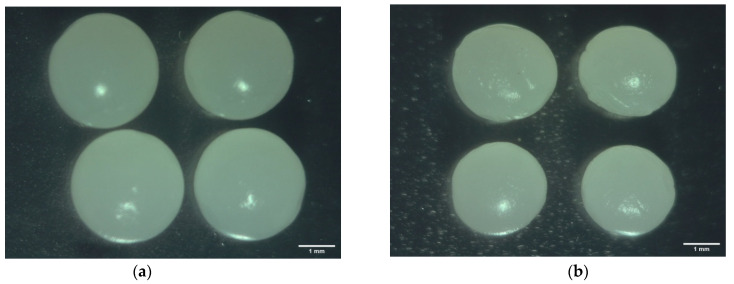
Microphotographs of calcium alginate-cornstarch hydrogel beads (alginate at 2% (*w*/*w*) and cornstarch at 2% (*w*/*w*)) filled with solution: (**a**) Initial solution; (**b**) Cycle 1; (**c**) Cycle 2; (**d**) Cycle 3.

**Table 1 polymers-14-01031-t001:** Experimental design (factors and levels for alginate and cornstarch solutions) ^1^.

Factors	Levels		
Alginate (%, *w*/*w*)	1	2	3
Cornstarch (%, *w*/*w*)	0.5	1	2

^1^ All treatments were performed with the initial model solution and each BFC cycle.

**Table 2 polymers-14-01031-t002:** EE% of the mixture solutions.

Alginate Concentration (%, *w*/*w*)	Cornstarch Concentration (%, *w*/*w*)	Solution Encapsulated	EE (%)
1.0	0.5	Initial solution	12.50 ± 1.45 ^a^
		Cycle 1	37.88 ± 0.94 ^c^
		Cycle 2	59.39 ± 1.33 ^d^
		Cycle 3	19.26 ± 2.65 ^b^
	1.0	Initial solution	21.25 ± 2.01 ^a^
		Cycle 1	60.84 ± 1.41 ^c^
		Cycle 2	64.62 ± 1.01 ^d^
		Cycle 3	22.06 ± 1.59 ^ab^
	2.0	Initial solution	25.83 ± 2.71 ^a^
		Cycle 1	64.42 ± 2.09 ^c^
		Cycle 2	69.02 ± 1.78 ^d^
		Cycle 3	32.21 ± 1.79 ^b^
2.0	0.5	Initial solution	47.67 ± 3.07 ^a^
		Cycle 1	75.91 ± 1.39 ^c^
		Cycle 2	82.92 ± 2.07 ^d^
		Cycle 3	50.74 ± 1.15 ^ab^
	1.0	Initial solution	52.27 ± 1.15 ^a^
		Cycle 1	83.50 ± 2.59 ^c^
		Cycle 2	87.48 ± 3.52 ^cd^
		Cycle 3	69.37 ± 2.11 ^b^
	2.0	Initial solution	63.28 ± 1.54 ^a^
		Cycle 1	90.17 ± 3.15 ^c^
		Cycle 2	97.73 ± 1.38 ^d^
		Cycle 3	75.09 ± 3.01 ^b^
3.0	0.5	Initial solution	38.95 ± 4.42 ^a^
		Cycle 1	52.12 ± 1.61 ^b^
		Cycle 2	64.90 ± 0.99 ^c^
		Cycle 3	40.01 ± 1.56 ^a^
	1.0	Initial solution	41.47 ± 2.13 ^a^
		Cycle 1	60.58 ± 2.71 ^c^
		Cycle 2	75.09 ± 2.11 ^d^
		Cycle 3	47.85 ± 3.01 ^b^
	2.0	Initial solution	57.22 ± 3.23 ^ab^
		Cycle 1	70.24 ± 4.66 ^c^
		Cycle 2	83.52 ± 4.23 ^d^
		Cycle 3	53.58 ± 1.85 ^a^

Different small letters indicate significant differences at *p* < 0.05, respectively.

**Table 3 polymers-14-01031-t003:** Characterization of calcium alginate hydrogel beads.

Sample	Particle Size (mm)	SF	a_w_	Moisture Content (%)	*ρ*_B_ (g/mL)
Initial solution	3.09 ± 0.02 ^a^	0.02 ± 0.002 ^a^	0.98 ± 0.01 ^a^	86.1 ± 0.15 ^a^	1.04 ± 0.05 ^a^
Cycle 1	2.82 ± 0.02 ^b^	0.02 ± 0.003 ^ab^	0.97 ± 0.00 ^ab^	81.5 ± 0.56 ^b^	1.05 ± 0.05 ^ab^
Cycle 2	2.73 ± 0.03 ^c^	0.02 ± 0.004 ^ab^	0.95 ± 0.01 ^c^	80.0 ± 0.10 ^bc^	1.11 ± 0.03 ^bc^
Cycle 3	2.64 ± 0.02 ^d^	0.02 ± 0.002 ^a^	0.92 ± 0.01 ^d^	73.6 ± 0.23 ^d^	1.16 ± 0.11 ^abcd^

^a–d^: Different superscripts within the same column indicate significant differences at *p* < 0.05.

## Data Availability

The data presented in this study are available on request from the corresponding author.

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
