# Peer review of "Optimization of Encapsulation by Ionic Gelation Technique of Cryoconcentrated Solution: A Response Surface Methodology and Evaluation of Physicochemical Characteristics Study"

_polymers, 2022, doi:10.3390/polym14051031_

Round 1

Reviewer 1 Report

The authors have studied the optimal conditions to obtain calcium alginate-cornstarch hydrogel beads filled with sucrose/acid gallic solution as cryoconcentrated solution. After preparation, the hydrogel beads were characterized in terms of efficiency of encapsulation, loading capacity, particles size and shape, water activity, moisture content and bulk density.

Finally, the authors found that a concentration of 2% (w/w) alginate and 2% (w/w) cornstarch mixture confers the optimum properties of the hydrogel beads, in order to be used for food, engineering, and/or pharmaceutical applications.

Some revisions are necessary:

1. At page 4, lines 166-167: Seems that the number of equation is 3 (not 1). Please correct it.

2. At pages 5-6, lines 204-219: For better view/ understanding, I suggest to the authors to include the values into a table.

3. At page 6, line 220: Please correct the last label “Efficiency of encapsulation: (a) Initial solution; (b) Cycle 1; (c) Cycle 2; (d) Cycle 3”.

4. At pages 5-7, section 3.1. “Efficiency of encapsulation (EE%)”: Please explain why the EE% values for cycle 3 (in all combinations) are smaller than for cycles 1 and 2.

5. At page 7, Figure 3: In this figure are visible only error bars, but not the columns. Please improve it.

6. At page 9, lines 350-352: Please correct the last label “(a) Initial solution; (b) Cycle 1; (c) Cycle 2; (d) Cycle 3”.

7. At pages 10-12, References: An unusual large number of self-citations are present for the author no.2 (10), Guillermo Petzold (23% of all references) and also for the author no.3 (9), Patricio Orellana-Palma (21% of all references). In order to be accepted for publishing, the authors must reduce the number of self-citations.

Author Response

RESPONSE TO REVIEWER 1

The authors have studied the optimal conditions to obtain calcium alginate-cornstarch hydrogel beads filled with sucrose/acid gallic solution as cryoconcentrated solution. After preparation, the hydrogel beads were characterized in terms of efficiency of encapsulation, loading capacity, particles size and shape, water activity, moisture content and bulk density.

Finally, the authors found that a concentration of 2% (w/w) alginate and 2% (w/w) cornstarch mixture confers the optimum properties of the hydrogel beads, in order to be used for food, engineering, and/or pharmaceutical applications.

Some revisions are necessary:

  1. At page 4, lines 166-167: Seems that the number of equation is 3 (not 1). Please correct it.

Thank you for the observation. We have corrected the lines. Please, see the subsection 2.7.2. Shape (red letters).

  1. At pages 5-6, lines 204-219: For better view/ understanding, I suggest to the authors to include the values into a table.

Thank for the suggestion. We have added a table for a better/view understanding of the lines 204-219. Please, see 3. Results and discussion, 3.1. Efficiency of encapsulation (EE%), Table 2, red letters.

  1. At page 6, line 220: Please correct the last label “Efficiency of encapsulation: (a) Initial solution; (b) Cycle 1; (c) Cycle 2; (d) Cycle 3”.

Thank you for the observation. We have corrected the label. Please, see the label of Figure 2 (red letters).

  1. At pages 5-7, section 3.1. “Efficiency of encapsulation (EE%)”: Please explain why the EE% values for cycle 3 (in all combinations) are smaller than for cycles 1 and 2.

Thank you for the suggestion. We have added information on the EE% values, where cycle 3 (in all combinations) are smaller than for cycles 1 and 2. Please, see 3. Results and discussion, 3.1. Efficiency of encapsulation (EE%), red letters.

  1. At page 7, Figure 3: In this figure are visible only error bars, but not the columns. Please improve it.

Thank you for the observation. We have arranged the Figure 3.

  1. At page 9, lines 350-352: Please correct the last label “(a) Initial solution; (b) Cycle 1; (c) Cycle 2; (d) Cycle 3”.

Thank you for the observation. We have corrected the label. Please, see the label of Figure 4 (red letters).

  1. At pages 10-12, References: An unusual large number of self-citations are present for the author no.2 (10), Guillermo Petzold (23% of all references) and also for the author no.3 (9), Patricio Orellana-Palma (21% of all references). In order to be accepted for publishing, the authors must reduce the number of self-citations.

Thank you for the suggestion. We have reduced the number of self-citations.

For the introduction, we have added references with similarities to the previous citations (Please see, References, number 3, 4, 8 and, 10, red letters), and we have eliminated a big percentage of self-citations.

For materials and methodology, the references were maintained (Number 22 and 5), since the present study comes from previous manuscripts (Doctoral Thesis: María Guerra-Valle).

For results and discussion, some references were maintained, since they are a fundamental part to compare some results. However, other references were removed, since the references were redundant in the manuscript (number 34 and 35).

Thus, the percentage of self-citation was reduced to 12% for Orellana-Palma and 12% for PEtzold.

Dr. Patricio Orellana-Palma              Dr. Guillermo Petzold

Depo de Ingeniería en Alimentos    Departamento de Ingeniería en Alimentos

Universidad de La Serena                 Universidad del Bío-Bío

Tel: +56-51-2204000                        Tel: +56-42-2463173

E-mail: [email protected]           E-mail: [email protected]

Reviewer 2 Report

I have read the manuscript “Optimization of encapsulation by ionic gelation technique of cryoconcentrated solution: A response surface methodology and evaluation of physicochemical characteristics study” by María Guerra-Valle et al. (MS # Polymers-1607397) submitted for the publication in Polymers.

In their short manuscript the authors reported the preparation and characterization (rather limited) of capsules containing cryo-concentrated solutions obtained by ionic gelation.  Results showed the existence of an optimal composition for the formation of the external shell.

As reported by the same authors in the discussion section, the use of alginate and corn-starch for the formation of capsules shell has been reported by several authors, who encapsulated different solutions but used more or less the same concentrations for alginate and corn-starch. Consequently, it is opinion of the referee that the manuscript need major revisions before its publication in Polymers. In particular:

  1. The size of capsules is reported in cm all over the text and in Table 2, but the scale bars in Figure 4 is in mm;
  2. Line 96: what is the rationale for an axial freezing? It could be the source of a gradient in the freezing profile;
  3. Lines 150-151 and 154: please, check the sentences;
  4. Line 166 Equation 1: it should be Equation 3;
  5. Line 175: something is missing;
  6. Figure 2: the particular angular representation does not give the real trends for EE%;
  7. Lines 234-239, line 249, and followings: several times the authors wrote/hypothesized about the formation of a dense shell, high solution viscosity and pores filled by corn-starch, but they did not give any measurements that could confirm such hypothesis such as a SEM picture, a release profile, or a rheological measurement;
  8. Figure 4: pictures give the idea of flattened capsules: is it really so? Cross pictures of the capsules are mandatory. If the capsules change their shape under gravity the SF values reported in Table 2 need a deeper discussion;
  9. Figure 4: the capsules look like to be quite different in size and not in agreement with data in Table 2.

Author Response

RESPONSE TO REVIEWER 2

I have read the manuscript “Optimization of encapsulation by ionic gelation technique of cryoconcentrated solution: A response surface methodology and evaluation of physicochemical characteristics study” by María Guerra-Valle et al. (MS # Polymers-1607397) submitted for the publication in Polymers.

In their short manuscript the authors reported the preparation and characterization (rather limited) of capsules containing cryoconcentrated solutions obtained by ionic gelation.  Results showed the existence of an optimal composition for the formation of the external shell.

As reported by the same authors in the discussion section, the use of alginate and corn-starch for the formation of capsules shell has been reported by several authors, who encapsulated different solutions but used more or less the same concentrations for alginate and corn-starch. Consequently, it is opinion of the referee that the manuscript need major revisions before its publication in Polymers. In particular:

  1. The size of capsules is reported in cm all over the text and in Table 2, but the scale bars in Figure 4 is in mm.

Thank you for the observation. We have corrected this point. Please, see abstract, Table 3 (formerly called Table 2. The change in the number of table is due to comments from reviewer 1), and 3.3. Characterization of calcium alginate hydrogel beads (red letters).

  1. Line 96: what is the rationale for an axial freezing? It could be the source of a gradient in the freezing profile.

Thank you for the question. The reason for an axial freezing is as follows. Firstly, the freeze concentrations is based on the movement of solutes during the formation of ice crystals during the freezing process and due to this process the cryoconcentrated solution is expelled and accumulated in the liquid fraction (between ice crystals). Therefore, it is important to control the freezing rate, since, when a freezing rate is higher than 8 μm/s, the freezing occurred too rapidly to expect a considerable separation of the concentrated solution from the ice crystals, since under these conditions (freezing rate > 8 μm/s) the ice occluded solutes during the freezing process. Thus, once the freeze concentration is finished, the final ice matrix can be visualized as a porous structure, where the cryoconcentrated solution moves to the outside through the channels formed by the ice crystals, and the separation process can be realized by gravitational thawing or by external forces (centrifugation or vacuum).

In previous studies on freeze concentration under different freezing conditions, axial freezing (freezing front from top to bottom) was compared with radial freezing (freezing front from the walls to the center), and the results showed that the axial freezing presented lower freezing rate (<8 μm/s) than radial freezing (>8 μm/s), since radial treatments have freezing front propagation in all the radial direction, and axial treatments have freezing front propagation in only one way direction axis.

                     Radial freezing                                 Axial freezing

Hence, the axial freezing process is more slowly, and it allows a moderate crystal growth with better separation of cryoconcentrate solution from the water (ice crystals), i.e., the movement by counter-diffusion of solutes away from the growing crystal surface and accumulate at the solid-liquid interface was better in axial freezing than radial freezing. Thereby, the separation process is easier, and in turn, the results have shown highest efficiency of separation in axial freezing.

For example, in sucrose solution (Petzold and Aguilera, 2013), the efficiency values for axial freezing were close to 75%,while for radial freezing, the efficiency values were close to 69%. In fruit juice (Orellana-Palma et al., 2017), axial freezing at -20 °C showed a significant solute increased in the concentrated fraction, with a value of 63 °Brix, equivalent to an increase of 3.8 times in the total polyphenol content (76% of retention), and the color of concentrated samples was darker than the initial sample, with ΔE* values of >25 CIELab units in all treatments.

Orellana-Palma, P., Petzold, G., Pierre, L. and Pensaben, J.M. (2017). Protection of polyphenols in blueberry juice by vacuum-assisted block freeze concentration. Food and Chemical Toxicology, 109, 1093-1102.

Petzold, G. and Aguilera, J.M. (2013). Centrifugal freeze concentration. Innovative Food Science and Emerging Technologies, 20, 253-258.

  1. Lines 150-151 and 154: please, check the sentences.

Thank you for the observation. The lines were revised and rephrased for a better understanding of the reader. Please, see 2.6.2. Loading capacity (LC%) (red letters).

  1. Line 166 Equation 1: it should be Equation 3.

Thank you for the observation. We have corrected the line. Please, see the subsection 2.7.2. Shape (red letters).

  1. Line 175: something is missing.

Thank you for the observation. We have corrected the line. Please, see the subsection 2.7.4. Moisture content (red letters).

  1. Figure 2: the particular angular representation does not give the real trends for EE%

Thank you for the point. To avoid confusion, we have improved the subsection 3.1 Efficiency of encapsulation (EE%) with comments from the reviewer 2 and reviewer 1. We have added a table for a better/view understanding of the Figure 2. In this way, the trend of Figure 2 is supported by Table 2. Please, see 3. Results and discussion, 3.1. Efficiency of encapsulation (EE%), Table 2 (red letters).

  1. Lines 234-239, line 249, and followings: several times the authors wrote/hypothesized about the formation of a dense shell, high solution viscosity and pores filled by corn-starch, but they did not give any measurements that could confirm such hypothesis such as a SEM picture, a release profile, or a rheological measurement;

Thank you for the point. Indeed, other studies indicate SEM images, release profiles (in different medium or in vitro process), fluorescence microscopy, or rheological measurements, among others. However, from December (from Christmas holidays), Chile has been negatively impacted by the COVID-19 pandemic, and thus, we were in quarantine. Specifically, from January, the numbers of infections per day increased significantly, and thus, the universities have decided to close until April. Hence, we cannot go to the University to work, and it is not possible to perform more results due to the multiple restrictions derived from the COVID-19 pandemic. Previously, we informed to the editorial office of Polymers on the situation, since the invitation to submit a manuscript was in December 2021, and from December, we were obtaining results, until the restrictions arrived. Additionally, we added the curve for COVID-19 infections in Chile.

Based on the above, we decided to carry out a first study on a model solution, to later carry out a more detailed study with real cryoconcentrated juice, with a greater amount of analysis on the encapsulation, process parameters, in different environments, and physicochemical parameters, among others.

  1. Figure 4: pictures give the idea of flattened capsules: is it really so? Cross pictures of the capsules are mandatory. If the capsules change their shape under gravity the SF values reported in Table 2 need a deeper discussion;

Thank you for the question and observation. The hydrogel capsules have a spherical shape. We understand your comment on flattened capsules, but it is an angle concept. We add two images from another angle of the capsules, where it is to observe their spherical shape.

The images will not be included in the manuscript, since they are images without quality requirements for the journal.

For Table 2, the results correspond to the best treatment, in our case alginate at 2% (w/w) and cornstarch at 2% (w/w), where the gravity does not affect the sphericity factor (no significant differences). In 3.1. Efficiency of encapsulation (EE%), the results correspond to a complete visualization of all the treatments in terms of efficiency, and thus, from the best results, we continue the manuscript. However, we have images of other treatments, where it is possible to observe that the capsule did not achieve a complete formation due to the low alginate concentration (alginate at ≤1% (w/w)), as well as, a high concentration of alginate (>2% (w/w)) generates "tails" in the capsules due to the effect of viscosity.

                                                                          Alginate at 1% (w/w)

                                                                          Cornstarch at 0.5% (w/w)

                                                                          Initial solution

                                                                          Incomplete formation of capsules

                                                                                      Alginate at 3% (w/w)

                                                                                      Cornstarch at 2% (w/w)

                                                                                     Cycle 3

                                                                                     Capsules with tails

  1. Figure 4: the capsules look like to be quite different in size and not in agreement with data in Table 2.

Thank you for your point. We have corrected the dimensions in the manuscript, since mm is the correct dimension. Please, see abstract, Table 3 (formerly called Table 2. The change in the number of table is due to comments from reviewer 1), and 3.3. Characterization of calcium alginate hydrogel beads (red letters).

Dr. Patricio Orellana-Palma                           Dr. Guillermo Petzold

Depto de Ingeniería en Alimentos               Depto de Ingeniería en Alimentos

Universidad de La Serena                             Universidad del Bío-Bío

Tel: +56-51-2204000                                     Tel: +56-42-2463173

E-mail: [email protected]           E-mail: [email protected]

Reviewer 3 Report

Optimization of encapsulation by ionic gelation technique of cryo concentrated solution: A response surface methodology and evaluation of physicochemical characteristics study deals with the optimal process conditionings. Design of experiments (DOE) was evaluated using surface response method and associated results are presented, however it needs revisions.

  1. The reason of chosen parameters for optimization should be discussed.
  2. Literature on surface response and DOE is too little, the reasons of using optimization are needed. Its strength in different fields should be discussed, followings can be added in terms process optimizations:
    1. Optimization of mechanical and thermal properties of iPP and LMPP blend fibres by surface response methodology
    2. Optimization of annealing stack using design of experiment method in Batch Annealed HSLA Steel
    3. Process optimization of eco-friendly flame retardant finish for cotton fabric: A response surface methodology approach
    4. Optimizing organophosphorus fire-resistant finish for cotton fabric using box-behnken design
  3. Is it possible to add more experimental results?
  4. The effects of the optimization should be shown, with effective compositions.
  5. The conclusion part should be concise to the main findings.

Author Response

RESPONSE TO REVIEWER 3

Optimization of encapsulation by ionic gelation technique of cryoconcentrated solution: A response surface methodology and evaluation of physicochemical characteristics study deals with the optimal process conditionings. Design of experiments (DOE) was evaluated using surface response method and associated results are presented, however it needs revisions.

  1. The reason of chosen parameters for optimization should be discussed.

The reason of chosen parameters for optimization are based on the fact that alginate is the most used material for encapsulation due to several advantages such as its non-toxicity, high biocompatibility and low cost (Burgain et al., 2011). However, alginate has low mechanical resistance, and thus, the beads present a high amount of pores, and it can generate inadequate protection of the encapsulating material. Hence, cornstarch was used as filled to avoid the output of encapsulated material, and thus, the calcium alginate-cornstarch hydrogel beads presented high efficiency of encapsulation (Rohman et al., 2021; Sultana et al., 2000).

Previous studies have indicated that concentrations of alginate solution between 0.5 to 4% (w/w) and cornstarch between 0.5 to 3% (w/w) can be considered suitable for encapsulating any liquid material, and in our case, as it is the first time in the encapsulation of a concentrated liquid obtained by freeze concentration, we decided to study the concentration of alginate between 1 to 3% (w/w) and cornstarch between 0.5 to 2% (w/w) (López-Córdoba et al., 2014), since studies indicate that a high concentrations of alginate, cornstarch and encapsulated material affect the final sphericity and efficiency of encapsulation of the capsules due to the high viscosity of the mixture (López-Córdoba et al., 2013).

Burgain, J., Gaiani, C., Linder, M. and Scher, J. (2011). Encapsulation of probiotic living cells: from laboratory scale to industrial applications. Journal of Food Engineering, 104, 467-483.

Sultana, K., Godward, G., Reynolds, N., Arumugaswamy, R., Peiris, P. and Kailasapathy, K. (2000). Encapsulation of probiotic bacteria with alginate–starch and evaluation of survival in simulated gastrointestinal conditions and in yoghurt. International Journal of Food Microbiology, 62, 47-55.

López-Córdoba, A., Deladino, L. and Martino, M. (2014). Release of yerba mate antioxidants from corn starch–alginate capsules as affected by structure. Carbohydrate Polymers, 99, 150-157.

López-Córdoba, A., Deladino, L. and Martino, M. (2013). Effect of starch filler on calcium-alginate hydrogels loaded with yerba mate antioxidants. Carbohydrate Polymers, 95, 315-323.

  1. Literature on surface response and DOE is too little, the reasons of using optimization are needed. Its strength in different fields should be discussed, followings can be added in terms process optimizations:
  2. Optimization of mechanical and thermal properties of iPP and LMPP blend fibres by surface response methodology
  3. Optimization of annealing stack using design of experiment method in Batch Annealed HSLA Steel
  4. Process optimization of eco-friendly flame retardant finish for cotton fabric: A response surface methodology approach
  5. Optimizing organophosphorus fire-resistant finish for cotton fabric using box-behnken design

Thank you for the recommendations. We have added information with references in the section of introduction. Please, see 1. Introduction (red letters) and References (red letters).

  1. Is it possible to add more experimental results?

Thank you very much for the question, however it is not possible to add more experimental results since there are still restrictions derived from the covid-19 pandemic for access to university laboratories. Specifically, from December (from Christmas holidays), Chile has been negatively impacted by the COVID-19 pandemic, and thus, we were in quarantine. Specifically, from January, the numbers of infections per day increased significantly, and thus, the universities have decided to close until April. Hence, we cannot go to the University to work, and it is not possible to perform more experimental results due to the multiple restrictions derived from the COVID-19 pandemic. Previously, we informed to the editorial office of Polymers on the situation, since the invitation to submit a manuscript was in December 2021, and from December, we were obtaining results, until the restrictions arrived. Additionally, we added the curve for COVID-19 infections in Chile.

Graphic Covid-19 Chile

Based on the above, we decided to carry out a first study on a model solution, to later carry out a more detailed study with real cryoconcentrated juice, with a greater amount of analysis on the encapsulation, process parameters, in different environments, and physicochemical parameters, among others.

  1. The effects of the optimization should be shown, with effective compositions.

Thank you for the point. To avoid confusion, we have improved the subsection 3.1 Efficiency of encapsulation (EE%) with comments from the reviewer 1, reviewer 2 and reviewer 3. We have added a table for a better/view understanding of the Figure 2. In this way, the trend of Figure 2 is supported by Table 2. Please, see 3. Results and discussion, 3.1. Efficiency of encapsulation (EE%), Table 2 (red letters).

  1. The conclusion part should be concise to the main findings.

Thank you very much for the recommendation. We have improved section of conclusions. Please, see 4. Conclusions (red letters).

Dr. Patricio Orellana-Palma                 Dr. Guillermo Petzold

Depto de Ingeniería en Alimentos      Departamento de Ingeniería en Alimentos

Universidad de La Serena                    Universidad del Bío-Bío

Tel: +56-51-2204000                           Tel: +56-42-2463173

E-mail: [email protected]  E-mail: [email protected]

Round 2

Reviewer 2 Report

The authors have answered to almost questions in a satisfactory manner, but I suggest them to add in the manuscript, before its pubblication in Polymers:

  1. a sentence regarding the advantages (demonstrated in previous works) of an axial freezing,
  2. a sentence regarding the agreement of their hypothesis of the formation of a dense shell and pores filled by corn-starch  with previous works present in literature, 
  3. the cross section of capsules in a Supplementary Information file even if they can be of low quality,
  4. again in Supplementary Information file, a Table which summarize the main results in terms of capsule quality that they wrote in their reply to referral report.

Author Response

RESPONSE TO REVIEWER 2

The authors have answered to almost questions in a satisfactory manner, but I suggest them to add in the manuscript, before its pubblication in Polymers:

  1. A sentence regarding the advantages (demonstrated in previous works) of an axial freezing.

Thank you for the suggestion. We have added the advantages of an axial freezing through previous studies. An important point, the reviewer 1 recommended reducing self-citations, and thus, we withdrew a considerable number of our publications, but we will add references from our group of studies that indicate the advantages of axial freezing. Please, see 2. Materials and Methods, 2.2. Preparation of cryoconcentrated solutions, red letters.

  1. A sentence regarding the agreement of their hypothesis of the formation of a dense shell and pores filled by corn-starch with previous works present in literature.

Thank you for the suggestion. We have added a sentence on the formation of a dense shell and pores filled by corn-starch with previous works present in literature. Please, see 3. Results and discussion, 3.1. Efficiency of encapsulation (EE%), red letters.

  1. The cross section of capsules in a Supplementary Information file even if they can be of low quality.

Thank me for the suggestion. However, we do not have images from cross-sections of the capsules, since from January (as mentioned), we have not been able to enter the University, and it is not advisable to send the Doctoral student to work, since the government has prohibited leaving our houses. First health, then work. We have elaborated a Supplementary Information file with the images that were previously shown (with more treatment interactions).

  1. Again in Supplementary Information file, a Table which summarize the main results in terms of capsule quality that they wrote in their reply to referral report.

Thank me for the suggestion. We have added a Table with main results in terms of capsule quality. An important point, Table 2. EE% of the mixture solutions corresponds to the main results in terms of efficiency, and it can be used as Supplementary Information file, but the reviewer 1 requested it in the manuscript.

Dr. Patricio Orellana-Palma               Dr. Guillermo Petzold

Depto de Ingeniería en Alimentos    Departamento de Ingeniería en Alimentos

Universidad de La Serena                  Universidad del Bío-Bío

Tel: +56-51-2204000                                     Tel: +56-42-2463173

E-mail: [email protected]           E-mail: [email protected]

Reviewer 3 Report

i have no more comments to add. the quality of the figures can be improved

Author Response

RESPONSE TO REVIEWER 3

I have no more comments to add. The quality of the figures can be improved.

Thank you for the comment. We will try to improve the quality of the figures.

Dr. Patricio Orellana-Palma                Dr. Guillermo Petzold

Depo de Ingeniería en Alimentos      Departamento de Ingeniería en Alimentos

Universidad de La Serena                   Universidad del Bío-Bío

Tel: +56-51-2204000                          Tel: +56-42-2463173

E-mail: [email protected]   E-mail: [email protected]